# Synthesis, Spectroscopic Characterization, Antibacterial Activity, and Computational Studies of Novel Pyridazinone Derivatives

**DOI:** 10.3390/molecules28020678

**Published:** 2023-01-09

**Authors:** Said Daoui, Şahin Direkel, Munjed M. Ibrahim, Burak Tüzün, Tarik Chelfi, Mohammed Al-Ghorbani, Mustapha Bouatia, Miloud El Karbane, Anass Doukkali, Noureddine Benchat, Khalid Karrouchi

**Affiliations:** 1Laboratory of Applied Chemistry and Environment (LCAE), Department of Chemistry, Faculty of Sciences, University Mohammed I, Oujda 60000, Morocco; 2Department of Medical Microbiology, Faculty of Medicine, Giresun University, Giresun 28100, Turkey; 3Department of Pharmaceutical Chemistry, College of Pharmacy, Umm Al-Qura University, Makkah 21955, Saudi Arabia; 4Science Faculty, Department of Chemistry, Cumhuriyet University, Sivas 58140, Turkey; 5Department of Chemistry, Ulla Science and Art College, Taibah University, KSA, Medina 42353, Saudi Arabia; 6Laboratory of Analytical Chemistry and Bromatology, Team of Formulation and Quality Control of Health Products, Faculty of Medicine and Pharmacy, Mohammed V University in Rabat, Rabat 10100, Morocco

**Keywords:** synthesis, pyridazinone, DFT, antibacterial, molecular docking, ADMET

## Abstract

In this work, a novel series of pyridazinone derivatives (**3**–**17**) were synthesized and characterized by NMR (^1^H and ^13^C), FT-IR spectroscopies, and ESI-MS methods. All synthesized compounds were screened for their antibacterial activities against *Staphylococcus aureus* (Methicillin-resistant), *Escherichia coli*, *Salmonella typhimurium*, *Pseudomonas aeruginosa*, and *Acinetobacter baumannii*. Among the series, compounds **7** and **13** were found to be active against *S. aureus* (MRSA), *P. aeruginosa*, and *A. baumannii* with the lowest MIC value range of 3.74–8.92 µM. Afterwards, DFT calculations of B3LYP/6-31++G(d,p) level were carried out to investigate geometry structures, frontier molecular orbital, molecular electrostatic potential maps, and gap energies of the synthesized compounds. In addition, the activities of these compounds against various bacterial proteins were compared with molecular-docking calculations. Finally, ADMET studies were performed to investigate the possibility of using of the target compounds as drugs.

## 1. Introduction

Infections with certain bacteria can cause serious diseases such as pneumonia, tuberculosis, and meningitis. These infections are considered one of the most common causes of persistent disease and affect millions of people with a very high level of morbidity and mortality [1]. Antibiotic drugs used in the treatment of bacterial infections have been used intensively, unconsciously, and excessively for many years, leading to the development of resistance and the spread of resistant pathogens. Thus, antibiotic-resistant bacteria are the main risk to human health and this resistance to antibiotic drugs has become a major concern for researchers [2]. There is therefore an urgent need to develop and identify new antibiotic agents to combat resistant pathogens with a new mode of action and broad-spectrum activities.

On other hand, the chemical and biological studies of pyridazine derivatives have been of great interest for many years for medicinal and agricultural reasons due to their large spectrum of biological properties including antimicrobial [3,4], antileishmania [5], antiviral [6], anticancer [7], anti-tubercular [8], analgesic and anti-inflammatory [9,10], anti-Alzheimer [11], antihypertensive [12], and anticonvulsant activities [13]. A substantial number of pyridazinones have been reported to possess a wide variety of agrochemicals such as pesticides [14,15,16,17] and recently, several pyridazinone derivatives have been reported as potent antibacterial agents (Figure 1, compound A) [18]. For instance, Abu-Hashem et al. reported a new series of pyridazine derivatives as potent antimicrobial agents (Figure 1, compound B) [19]. Interestingly, some pyridazinone-based drugs such as zardaverine, emorfazone, imazodan, and levosimendan are already sold in clinical markets (Figure 1).

Theoretical calculations are among the popular methods of today with computational calculations becoming both more practical and faster as they have been greatly influenced by the developing technology. It has become crucial to know the influence of various groups on the chemical structures in order to understand the links of these groups with their biological properties [20,21,22,23,24].

Inspired with the aforementioned biological importance of these derivatives and in resumption of our ongoing programs directed toward the development of novel heterocyclic compounds that are potent therapeutic agents [25,26,27,28,29], we report herein the synthesis, characterization, and antibacterial activities of a novel series of pyridazinone derivatives. Firstly, the electronic and geometric characteristics of the synthesized compounds were investigated by using DFT calculations with B3LYP/6-31++G(d,p) level. Afterwards, their biological properties against various bacterial proteins such as the crystal structure of *Staphylococcus aureus* (PDB ID: 1JIJ), the crystal structure of *Pseudomonas aeruginosa* PAO1 (PDB ID: 2UV0), and the crystal structure of *Escherichia coli* K-12 (PDB ID: 4WUB) were investigated. Thus, the newly synthesized compounds (**3**–**17**) were evaluated in vitro for their antibacterial activities against *S. aureus* (methicillin resistant), *Escherichia coli*, *A. baumannii*, *P. aeruginosa*, and *S. typhimurium* bacterial strains. In addition, the possibility of being used as a drug was examined by examining the drug properties of bioactive molecules using ADMET analysis.

## 2. Results and Discussion

### 2.1. Chemistry

The synthesis of compounds (**3**–**17**) was carried out following to the steps shown in Figure 1. Pyridazin-3(2*H*)-one (**1**), which served as a starting material in this study, was prepared via a one-step synthetic route from 4-oxo-4-phenylbutanoic acid and hydrazine hydrate in accordance with the published procedure [30]. Subsequently, compounds (**3**–**7**) were obtained by the condensation reaction of pyridazin-3(2*H*)-one (**1**) with aromatic aldehydes **2a**–**e** [31]. Then, compounds (**3**–**7**) were condensed with ethyl bromoacetate in refluxing ethanol in the presence of sodium methoxide as base to afforded esters (**8**–**12**) in 71–92% yields [32]. Finally, the esters (**8**–**12**) were converted to the corresponding acids (**13**–**17**) by treating it with sodium hydroxide in dry ethanol, followed by in situ acidification with diluted hydrochloric acid [33,34]. The target pyridazin-3(2*H*)-one derivatives (**3**–**17**) (Table 1) were obtained and the molecular structure were confirmed by FT-IR, ^1^H NMR, ^13^C NMR, and ESI-MS (FT-IR, NMR and ESI-MS spectra are given in the SI).

### 2.2. Antibacterial Activity

The title compounds (**3**–**17**) were screened for their antibacterial activities against Gram-positive isolate (*S. aureus* (MRSA)), Gram-negative isolates (*E. coli*, *S. typhimurium*), and Gram-negative nonfermenter isolates (*P. aeruginosa, A. baumannii*). If the color of the alamarBlue (indicator dye) in the wells turned pink, bacterial growth was interpreted as continuing. If there was no live cell presence, no color change occurred. In the test, the last well that did not change from blue to pink was accepted as the minimal inhibition concentration (MIC) value. After evaluation, samples in the wells were inoculated on blood-agar medium and checked the next day. Interpretation of MICs of standard bacterial isolates compared to standard “Amikacin” were done according to Clinical and Laboratory Standards Institute (CLSI) criteria. The MIC values of the compounds (**3**–**17**) are shown in Table 2. Antibacterial activities of compound **13** was shown in Figure 2. In this study, negative and positive controls were evaluated in the 7th and 8th well, respectively (Figure 2).

As noted in Table 2, the MIC values of the tested compounds (**3**–**17**) indicated that most compounds exhibited moderate to significant activity (MIC = 3.74–36.21 µM) in comparison to the reference drug “Amikacin”. According to these results, some compounds (**3, 10, 11, 14, 15**, and **17**) were found to have antibacterial activities against at least two bacteria, albeit at high concentrations. Compounds **4, 5, 6, 9**, and **12** showed poor or no activity at concentrations studied against any of the bacteria. Two compounds, **7** and **13**, were found effective against all studied bacteria. Indeed, Compound **7** showed significant activity with MIC value 7.8 μM against *E.coli*, *S. aureus* (MRSA), *S. typhimurium*, and *A. baumannii*. In fact, compound **13** was found to exhibit the most potent in vitro antibacterial activity against Gram-negative bacteria, with MICs of 3.74 and 7.48 μM against *A. baumannii* and *P. aeruginosa*, respectively. Compound **3** was the most active compound against Gram-positive bacteria with an MIC value of 4.52 μM against *S. aureus* (MRSA).

In order to perform a structure–activity relationship (SAR) study, the tested compounds were divided into three series: the first series contained 4-(aryl)-6-phenylpyridazin-3(2*H*)-one (**3**–**7**), with different substitutions (CH_3_, Cl, NO_2_, and F) on the aryl ring; the second series of esters (**8**–**12**) resulted from the N-alkylation of compounds (**3**–**7**) by introducing the ethyl ester group; and the third series of acids (**13**–**17**) resulted from a hydrolysis reaction of esters (**8**–**12**). It was observed from the results of antibacterial activity that the compound with the fluro group at para position (compound **7**) in its molecular structure was more active against Gram-negative bacteria as compared to other derivatives in the first series, which may have been due the fact that F group may have been involved in the binding of the ligand with the receptor site of the bacteria. It can also be seen from the antibacterial activity data that when we replaced the F group with electron-donating CH_3_ group (compound **3**), an abrupt rise in activity was observed against the Gram-positive bacterium (*S. aureus* (MRSA)). Thus, introduction of the ethyle ester group at the N-1 position of pyridazin-3(2*H*)-one ring (compounds **8** and **12**) led to a fall in antibacterial activity of the synthesized pyridazin-3(2*H*)-one derivatives (compounds **3** and **7**). On the other hand, the hydrolysis of the ester led to an increase in the antibacterial activity of the synthesized derivatives (compounds **13** and **15**) against Gram-negative bacteria (*P. aeruginosa* and *A. baumannii*).

### 2.3. Theoretical Calculations

The theoretical calculations were compared with both the chemical properties of the molecules and their biological properties. It provided important information before many experimental studies were carried out [35]. The calculated quantum chemical parameters of compounds (**3-17**) using B3LYP/6-31++G(d,p) Level are given in Appendix A (from Appendix A). There are important advantages to knowing the active sites of the molecules and the influence of each group on their biological properties in order to synthesize new molecules that are more effective and more active. Thus, numerous quantum parameters have been calculated in the Gaussian calculations; among the most important of these are HOMO (highest occupied molecular orbital), which shows the ability of molecules to donate electrons [36], and LUMO (lowest unoccupied molecular orbital) which shows the ability to accept electrons of molecules [37]. The optimized structure, HOMO, LUMO and molecular electrostatic potential surface of all molecules (**3**–**17**) are given in Appendix A (from Appendix A). The optimized shapes of representative compounds **7** and **13** are given in Figure 3. The second and third images are HOMO and LUMO. In the last image, the electrostatic potentials of the structures are given. In the molecule, there are red-colored regions that are electron-rich regions and blue-colored regions that are electron-poor. These two regions are the active regions of the molecule for both accepting and donating electrons [38]. The computed quantum chemical descriptors based upon DFT calculations of representative compounds **7** and **13** are presented in Table 3.

In general, it is known that when the value of the HOMO parameter of the molecules is the most positive and the value of the LUMO parameter is the most negative, the activity is the highest [35]. Considering the above explanations, if HOMO is the most positive and LUMO the most negative, it will be the least among them. This will increase the activity of the molecules. As a result of the calculations, many parameters have been calculated and since each calculated parameter explains a different feature, each has a different importance. Among the calculated parameters, it is seen that the molecule with the most negative numerical value of the HOMO parameter of the molecules has higher activity at B3LYP level; compound **13** has higher activity than other molecules. Among these parameters, the ∆E energy gap value is another parameter that shows the activity of the molecule. Molecules with the lowest ∆E energy gap have the highest activities. Herein, no direct correlation with activity can be highlighted. One of the other important parameters calculated is electronegativity, which shows the ability of atoms in the molecule to attract bond electrons [36]. The higher the numerical value of this parameter, the more the atoms in the molecule will attract the bond electrons, which will decrease the activity of the molecule [37]. Herein is verified a direct correlation between electronegativity and antibacterial activity, it is seen that compound **13** has higher activity than other molecules. Hence, the bioactivity of this compound may be explained by the ability of the biological target to receive electrons, which may be important for stabilization of the active site. Two other important parameters are chemical hardness and its opposite, which show the polarization and softness properties of molecules, respectively. Thus, hard molecules are less reactive than soft molecules because they cannot easily give electrons to an acceptor [39]. In this case, no direct correlation with the activity of these compounds and these two calculated parameters can be highlighted. Although many parameters have been found as a result of theoretical calculations, very few of these parameters have figural representations.

After the Gaussian calculations, the activities of the title molecules against bacterial proteins were investigated. The activities of molecules were determined based on the chemical interactions that occur between the molecules and proteins These interactions, are hydrogen bonds, π-π bonds, and polar and hydrophobic interactions [40,41,42]. The binding parameters for all compounds (**3**–**17**) are given in Appendix A (from Appendix A). As a result of docking studies, the interactions of the proteins and the active sites of compounds **7** and **13** were determined (Figure 4). The binding parameters for **7** and **13** on *S. aureus* (PDB ID: 1JIJ), *P. aeruginosa PAO1* (PDB ID: 2UV0), and *E. coli K-12* (PDB ID: 4WUB) proteins are given in Table 4. These parameters give the value of the chemical interactions (glide hbond, glide evdw, and glide ecoul) and the values obtained about the poses (glide emodel, glide energy, glide einternal, and glide posenum) obtained from the interaction of the title ligands with the studied proteins [40,41].

Docking results against *S. aureus* (PDB ID: 1JIJ) protein indicated a well-conserved binding region but with slightly different predicted best binding-energy values. The best free-binding energy was found for compound **13** (−7.31 kcal/mol), followed by compound (−7.12 kcal/mol), in comparison to the other investigated compounds; amikacin (reference molecule) displayed favorable binding energy value (Table 4). The observed trend in the binding free-energy was found to be consistent with the experimental activity trend for the title molecules. When the interactions between proteins and molecules are examined, in Figure 4a, it is seen that the carbonyl oxygen in the carboxylic acid group attached to the pyridazine ring forms hydrogen bonds with the LYS 84 and HIE 50 proteins. Figure 4a shows the 2D interaction diagram of the 2UV0-**13** docked structure. In comparison to the other compounds studied, compound **13** showed the best binding free-energy (−5.69 kcal/mol). As shown in Figure 4b, it is seen that the carbonyl oxygen in the same group and the oxygen atom attached to it form a hydrogen bond with LYS 97 and a Pi-cation bond with ARG 96. Thus, compound **7** showed good least binding energy of −5.74 kcal/mol against *E. coli* 4WUB protein. As shown in Figure 4c, it is seen that the benzene ring on one side of the molecule makes a Pi−Pi stacking interaction with the HIE 83 protein. The observed trend in the binding free energy was found to be consistent with the experimental activity trend for the title molecules.

Analyzing the physicochemical properties of the developed drug hits is a crucial step in analyzing and determining their drug-likeness potential (Appendix A, from Appendix A). For this reason, we evaluated the drug-likeness potential of compounds **7** and **13** through computing various descriptors, such as molar mass of molecules (mol_MW), dipole moment (dipole), total solvent accessible surface area (SASA), volume, donorHB (given hydrogen bond), accptHB (accepted hydrogen bond), globularity descriptor (glob), predicted polarizability (QPpolrz), brain−blood (QPPMDCK) and intestinal−blood (QPPCaco) barriers of molecules, predicted skin permeability (QPlogKp), and number of likely metabolic reactions (#metab) (Table 5) [43,44,45] according to the Lipinski’s rule of five [46,47] and Jorgensen’s rule of three [48]. Molecular weights of the title compounds ranged from 485.58 to 564.48 Da, with compounds **7** and **13** having molecular weights less than 500 Da and obeying the first Lipinski rule for effective and safe drug delivery. Lipinski’s second rule stipulates that drug-like compounds should not possess more than five hydrogen-bond-donating groups. The two title compounds comply with this rule. Hydrogen-bond-accepting groups in the compounds **7** and **13** are 2 and 4, respectively, thus also meeting Lipinski’s third rule. Apart from all these above-mentioned parameters, two important parameters are violations of Lipinski’s rule of five (RuleOfFive) [46,47] and violations of Jorgensen’s rule of three (RuleOfThree) [48]. The numerical value of this important parameter is required to be zero, which is consistent with the values found for both compounds (Table 5). Looking at all the calculated parameter values, we find that these values are within the recommended range and it can be concluded that the title molecules had drugability characteristics according to Lipinski’s rules. Thus, it can be seen that there is no harm in the application of all molecules to human metabolism as a theoretical drug.

## 3. Materials and Methods

### 3.1. General Methods

TLC was used to follow the reactions using aluminum sheets with silica gel 60 F254. Buchi−Tottoli apparatus was used to measure the melting points. The infrared spectra were recorded by using Perkin-Elmer FT Pargamon 1000 PC Spectrophotometer (Cleveland, OH, USA) covering field 400–4000 cm^−1^. ^1^H NMR and ^13^C NMR spectra were recorded in DMSO-*d_6_* on JNM-ECZ500R/S1 FT NMR SYSTEM (JEOL) (500 MHz). High resolution mass spectrometry (HRMS) spectra were collected by using a Q Extractive Thermofischer Scientific ion trap spectrometer by using ESI ionization (Waltham, MA, USA).

### 3.2. Chemistry

#### 3.2.1. General Procedure for the Synthesis of Compounds (**3**–**7**)

To a mixture of pyridazin-3(2H)-one (**1**) (1 mmol) and sodium methanoate (0.065 g, 1.2 mmol) in 30 mL of dry ethanol, aromatic aldehyde (**2**) (1 mmol) was added drop-wise. The mixture was refluxed for 6 h, then left under stirring overnight at room temperature. After cooling, the mixture was acidified with concentrated HCl. The precipitate formed was filtered, washed with water, and recrystallized from ethanol to afford the pure products (**3**–**7**).

*4-(4-methylbenzyl)-6-phenylpyridazin-3(2H)-one* (**3**): White solid, Yield = 78%, m.p = 276 °C, FT-IR (ATR, cm^−1^): 3304 (NH), 3130–2846 (CH), 1647 (C=O), 1600 (N=C); ^1^H NMR (500 MHz, DMSO-*d*_6_) δ: 2.40 (s, 3H, CH_3_), 2.96 (s, 2H, CH_2_-Ar), 7.32–7.50 (m, 7H, H-Ar), 7.70 (d, 2H, *J* = 8.4 Hz, 2H, H-Ar), 7.88 (s, 1H, CH-pyr), 10.82 (s, 1H, CONH); ESI-MS: calculated for C_18_H_16_N_2_O [M + H]^+^: 275.1300, found: 275.0509.

*4-(4-chlorobenzyl)-6-phenylpyridazin-3(2H)-one* (**4**): White solid, Yield = 72%; m.p = 296 °C; FT-IR (ATR, cm^−1^): 3304 (NH), 3124–2893 (CH), 1647 (C=O), 1600 (N=C); ^1^H NMR (500 MHz, DMSO-*d*_6_, δ (ppm)) δ: 2.96 (s, 2H, CH_2_-Ar), 7.38–7.45 (m, 3H, H-Ar), 7.61 (d, 2H, *J* = 8.4 Hz, H-Ar), 7.70 (d, 2H, *J* = 8.4 Hz, H-Ar), 7.80 (d, 2H, *J* = 8.4 Hz, H-Ar), 7.90 (s, 1H, CH-pyri), 10.84 (s, 1H, CONH); ESI-HRMS: calculated for C_17_H_13_ClN_2_O [M + H]^+^: 297.0715, found: 297.1030.

*4-(2,6-dichlorobenzyl)-6-phenylpyridazin-3(2H)-one* (**5**): Brown solid, Yield = 94%, mp = 165 °C, FT-IR (ATR, cm^−1^): 3206 (NH), 3061–2875 (CH), 1646 (C=O), 1605 (C=N); ^1^H NMR (500 MHz, DMSO-*d*_6_, δ (ppm)) δ: 7.55–7.53 (m, 2H, H-Ar), 7.53 (d, *J* = 2.1 Hz, 2H, H-Ar), 7.41–7.35 (m, 4H, H-Ar), 6.89 (s, 1H, H-Pyz),4.10 (s, 2H, NCH_2_CO); ^13^C NMR (101 MHz, DMSO-*d*_6_, δ (ppm)) δ: 160.77, 144.32, 140.31, 135.15, 133.23, 130.63, 129.77, 129.56, 129.44, 129.32, 126.06, 125.95, 31.44. ESI-HRMS: calculated for C_17_H_12_Cl_2_N_2_O [M + H]^+^: 331.0350, found: 331.0020.

*4-(2-nitrobenzyl)-6-phenylpyridazin-3(2H)-one* (**6**): Brown solid, Yield = 96%; mp = 170 °C, FT-IR (ATR, cm^−1^): 3200 (NH), 3043–2865 (CH), 1646 (C=O), 1595 (C=N); ^1^H NMR (500 MHz, DMSO-*d*_6_, δ (ppm)) δ: 8.21 (d, *J* = 2.0 Hz, 1H, H-Ar), 8.05 (ddd, *J* = 8.3, 2.4, 1.0 Hz, 1H, H-Ar), 8.03 (s,1H, H-pyr), 7.81–7.78 (m, 3H, H-Ar), 7.56 (t, *J* = 7.9 Hz, 1H, H-Ar), 7.46–7.42 (m, 2H, H-Ar), 7.41–7.37 (m, 1H, H-Ar), 3.97 (s, 2H, CH_2_-Ar); ^13^C NMR (101 MHz, DMSO-*d*_6_, δ (ppm)) δ: 161.09, 148.32, 144.64, 141.94, 141.01, 136.41, 135.28, 130.36, 129.72, 129.42, 129.36, 126.21, 124.17, 122.10, 35.40. HRMS: calculated for C_17_H_13_FN_2_O [M − H]^−^: 306.0315, found: 306.0475.

*4-(4-fluorobenzyl)-6-phenylpyridazin-3(2H)-one* (**7**): White solid, Yield = 58%, m.p = 280 °C, FT-IR (ATR, cm^−1^): 3296 (NH), 1740 (C=O), 1652 (N=C); ^1^H NMR (500 MHz, DMSO-*d*_6_) δ: 2.94 (s, 2H, CH_2_-Ar), 7.17–7.20 (m, 2H, H-Ar), 7.38–7.44 (m, 3H, H-Ar), 7.70 (d, 2H, J = 8.4 Hz, H-Ar), 7.78 (m, 2H, H-Ar), 7.88 (s, 1H, H-pyri), 10.82 (s, 1H, CONH); ^13^C NMR (101 MHz, DMSO-*d*_6_, δ (ppm)) δ: 161.84, 158.31, 144.20, 142.41, 135.90, 135.68, 132.94, 129.66, 129.85, 129.51, 129.54, 129.51, 127.57, 35.31. ESI-HRMS: calculated for C_17_H_13_N_3_O_3_ [M + H]^+^: 281.1018, found: 281.3045.

#### 3.2.2. General Procedure for the Synthesis of Compounds (**8**–**12**)

Ethyl bromoacetate (0.53, 3.2 mmol) was added drop-wise to a solution of compounds (**3**–**7**) (3 mmol), K_2_CO_3_ (1.24 g, 9 mmol), and tetra *n*-butylammonium bromide (TBAB) as a catalyst in dry THF (20 mL), and the mixture was refluxed for 6 h. After the end of the reaction, the formed salts were removed by filtration and the solvent was evaporated under reduced pressure. The residue was purified by chromatography on a column of silica gel (ethylacetate:hexane (1:1)) to give pure esters (**8**–**12**).

*Ethyl 2-(5-(4-methylbenzyl)-6-oxo-3-phenylpyridazin-1(6H)-yl)acetate* (**8**): White crystals, Yield = 71%, m.p = 120 °C, FT-IR (ATR, cm^−1^): 3489 (OH), 3078–2852 (CH), 1751, 1652 (C=O), 1607 (C=N); ^1^H NMR (500 MHz, DMSO-*d*_6_, δ (ppm)) δ: 7.91 (s, 1H,CH-pyri), 7.82 (d, *J* = 7.8 Hz, 2H, H-Ar), 7.51–7.43 (m, 3H, H-Ar) 7.24 (d, *J* = 7.9 Hz, 2H, H-Ar), 7.11 (d, *J* = 7.8 Hz, 2H, 2H, H-Ar), 4.94 (s, 2H, NCH_2_CO), 4.16 (q, *J* = 7.1 Hz, 2H,CH_2_CH_3_), 3.87 (s, 2H,CH_2_-Ar), 1.19 (t, *J* = 7.1 Hz, 3H, OCH_2_CH_3_). ^13^C NMR (101 MHz, DMSO-*d*_6_, δ (ppm)) δ: 167.86, 159.48, 143.74, 142.56, 135.50, 134.69, 134.24, 129.40, 128.97, 128.85, 128.03, 125.79, 61.10, 53.99, 35.03, 20.57, 13.91. HRMS: calculated for C_22_H_22_N_2_O_3_ [M + H]^+^: 363.1794, found: 363.1689.

*Ethyl 2-(5-(4-chlorobenzyl)-6-oxo-3-phenylpyridazin-1(6H)-yl)acetate* (**9**): White solid, Yield = 92%; m.p = 142 °C, FT-IR (ATR, cm^−1^): 3466 (OH), 3067–2864 (CH), 1745, 1647 (C=O), 1607 (C=N); ^1^H NMR (500 MHz, DMSO-*d*_6_, δ (ppm)) δ: 8.02 (s, 1H,CH-pyri), 7.85 (d, *J* = 7.8 Hz, 2H, H-Ar), 7.54–7.44 (m, 3H, H-Ar), 7.40 (d, *J* = 8.7 Hz, 2H, H-Ar), 7.35 (d, *J* = 8.6 Hz, 2H, H-Ar),7.43–7.31 (m, 4H, H-Ar), 4.96 (s, 2H, NCH_2_CO), 4.16 (q, *J* = 7.1 Hz, 2H, OCH_2_CH_3_), 3.92 (s, 2H, CH_2_-Ar), 1.19 (t, *J* = 7.1 Hz, 3H, CH_2_CH_3_); ^13^C NMR (101 MHz, DMSO-*d*_6_, δ (ppm)) δ: 167.48, 159.29, 143.74, 141.76, 136.85, 134.21, 131.21, 130.83, 129.41, 128.84, 128.44, 128.29, 125.82, 61.08, 53.82, 34.86, 13.91. HRMS: calculated for C_21_H_19_ClN_2_O_3_ [M + 3H]^+^: 385.1166, found: 385.1512.

*Ethyl 2-(5-(2,6-dichlorobenzyl)-6-oxo-3-phenylpyridazin-1(6H)-yl)acetate* (**10**): White solid, Yield = 78%; m.p = 80 °C, FT-IR (ATR, cm^−1^): 3304 (OH), 3067–2870 (CH), 1757, 1659 (C=O), 1612 (C=N); ^1^H NMR (500 MHz, DMSO-*d*_6_, δ (ppm)) δ: 7.58–7.55 (m, 2H, H-Ar), 7.49 (d, *J* = 8.1 Hz, 2H, H-Ar), 7.38–7.32 (m, 4H, H-Ar), 6.97 (s, 1H, CH-pyri), 5.02 (s, 1H, NCH_2_CO), 4.28 (s, 1H, CH_2_-Ar), 4.24 (q, *J* = 7.0 Hz), 1.27 (t, *J* = 7.1 Hz); ^13^C NMR (101 MHz, DMSO-*d*_6_, δ (ppm)) δ: 167.84, 160.38, 145.27, 140.10, 136.03, 134.57, 133.05, 129.60, 129.39, 128.66, 128.55, 126.31, 125.71, 61.58, 54.03, 31.00, 13.10. HRMS: calculated for C_21_H_18_Cl_2_N_2_O_3_ [M + H]^+^: 417.0796, found: 417.0757.

*Ethyl 2-(5-(2-nitrobenzyl)-6-oxo-3-phenylpyridazin-1(6H)-yl)acetate* (**11**): Brown solid, Yield = 85%; m.p = 175 °C, FT-IR (ATR, cm^−1^): 3211–2887 (CH), 1647 (C=O), 1607 (C=N); ^1^H NMR (500 MHz, DMSO-*d*_6_, δ (ppm)) δ: 8.22 (t, *J* = 1.9 Hz, 1H, H-Ar), 8.14 (s, CH-pyri), 8.06 (ddd, *J* = 8.2, 2.4, 1.0 Hz, 1H, H-Ar), 7.84–7.81 (m, 2H, H-Ar), 7.80 (dt, *J* = 7.7, 1.2 Hz, 1H, H-Ar), 7.56 (t, *J* = 8.0 Hz, 1H, H-Ar), 7.49–7.41 (m, 3H, H-Ar), 4.90 (s, 2H, NCH_2_CO), 4.09 (q, *J* = 7.1 Hz, 2H, OCH_2_CH_3_), 4.02 (s, 2H, CH_2_-Ar), 1.12 (t, *J* = 7.1 Hz, 3H, CH_2_CH_3_). ^13^C NMR (101 MHz, DMSO-*d*_6_, δ (ppm)) δ: 168.06, 159.82, 148.33, 144.37, 141.58, 140.73, 136.46, 134.71, 130.37, 130.10, 129.61, 129.48, 126.46, 124.28, 122.18, 61.65, 54.44, 35.72, 14.47. HRMS: calculated for C_21_H_18_N_3_O_4_ [M + 2H]^+^: 395.1307, found: 395.1144.

*Ethyl 2-(5-(4-fluorobenzyl)-6-oxo-3-phenylpyridazin-1(6H)-yl)acetate* (**12**): White solid, Yield = 88%, m.p = 73 °C, FT-IR (ATR, cm^−1^): 3484 (OH), 3067–2835 (CH), 1757, 1652 (C=O), 1600 (C=N); ^1^H NMR (500 MHz, DMSO-*d*_6_, δ (ppm)) δ: 7.58–7.54 (m, 4H, H-Ar), 7.45–7.35 (m, 5H, H-Ar), 6.93 (s, 1H, CH-pyri), 4.90 (s, 2H, NCH_2_CO), 4.10 (q, *J* = 7.1 Hz, 2H, CH_2_CH_3_), 3.87 (s, 2H, CH_2_-Ar), 1.14 (t, *J* = 7.1 Hz, 3H, OCH_2_CH_3_). ^13^C NMR (101 MHz, DMSO-*d*_6_, δ (ppm)) δ: 166.46, 159.57, 143.65, 139.80, 135.91, 134.84, 130.68, 129.94, 129.55, 129.36, 126.26, 125.76, 61.64, 54.40, 35.99, 14.50. HRMS: calculated for C_21_H_18_FN_2_O_·_ [M + H]^+^: 367.1435, found: 367.1427.

#### 3.2.3. General Procedure for the Synthesis of Compounds (**13**–**17**)

To a solution of esters (**8**–**12**) (3.6 mmol) in ethanol (50 mL), was added 6 N NaOH (14.4 mmol) and the mixture was stirred at room temperature for 4 h. The solvent was then evaporated and the residue was diluted with cold water, and acidified with 6 N HCl. The final product was filtered off by suction filtration and recrystallized from dry ethanol. The precipitate formed was filtered off and recrystallized from dry ethanol to give the corresponding acids (**13**–**17**).

*2-(5-(4-methylbenzyl)-6-oxo-3-phenylpyridazin-1(6H)-yl)acetic acid* (**13**): White solid, Yield = 90%, m.p = 170 °C, FT-IR (ATR, cm^−1^): 3461 (OH), 3072–2843 (CH), 1757, 1734 (C=O), 1654 (C=N); ^1^H NMR (500 MHz, DMSO-*d*_6_, δ (ppm)) δ: 7.91 (s, 1H, CH-pyri), 7.82 (d, *J* = 7.1 Hz, 2H, H-Ar), 7.54–7.43 (m, 3H, H-Ar), 7.25 (d, *J* = 7.7 Hz, 2H, H-Ar), 7.12 (d, *J* = 7.7 Hz, 2H, H-Ar), 4.87 (s, 2H, NCH_2_CO), 3.87 (s, 2H, CH_2_-Ar), 2.27 (s, 3H,CH_3_). ^13^C NMR (101 MHz, DMSO-*d*_6_, δ (ppm)) δ: 168.95, 159.37, 143.47, 142.53, 135.47, 134.76, 134.37, 129.34, 128.99, 128.90, 128.86, 127.87, 125.78, 53.73, 35.07, 20.62; HRMS: calculated for C_19_H_18_N_2_O_3_ [M + H]^+^: 335.1160, found: 335.1378.

*2-(5-(4-chlorobenzyl)-6-oxo-3-phenylpyridazin-1(6H)-yl)acetic acid* (**14**): White solid, Yield = 84%, m.p = 174 °C (EtOH), FT-IR (ATR, cm^−1^): 3461 (OH), 3055–2852 (CH), 1751, 1647 (C=O), 1600 (C=N); ^1^H NMR (500 MHz, DMSO-*d*_6_, δ (ppm)) δ: 7.99 (s, 1H, CH-pyri), 7.84 (dd, *J* = 7.8, 1.6 Hz, 2H, H-Ar), 7.54–7.44 (m, 3H, H-Ar), 7.41 (d, *J* = 8.7 Hz, 2H, H-Ar), 7.36 (d, *J* = 8.7 Hz, 2H, H-Ar), 4.81 (s, 2H, NCH_2_CO), 3.71 (s, 2H, CH_2_-Ar); ^13^C NMR (101 MHz, DMSO-*d*_6_, δ (ppm)) δ: 169.23, 159.29, 143.28, 141.66, 137.02, 134.41, 131.13, 130.87, 129.31, 128.86, 128.31, 128.14, 125.79, 54.15, 34.88. HRMS: calculated for C_19_H_15_ClN_2_O_3_ [M + H]^+^: 354.9932, found: 354.9837.

*2-(5-(2,6-dichlorobenzyl)-6-oxo-3-phenylpyridazin-1(6H)-yl)acetic acid* (**15**): White solid, Yield = 89%; m.p = 140 °C, FT-IR (ATR, cm^−1^): 3461 (OH), 3055–2852 (CH), 1751, 1647 (C=O), 1600 (C=N); ^1^H NMR (500 MHz, DMSO-*d*_6_, δ (ppm)) δ: 7.65–7.60 (m, 2H, H-Ar), 7.57 (d, *J* = 7.8 Hz, 2H, H-Ar), 7.48–7.40 (m, 4H, H-Ar), 7.03 (s, 1H, CH-pyri), 4.95 (s, 1H, NCH_2_CO), 4.19 (s, 1H, CH_2_-Ar); ^13^C NMR (101 MHz, DMSO-*d*_6_, δ (ppm)) δ: 168.89, 158.91, 143.24, 139.37, 135.34, 134.10, 132.49, 130.06, 129.43, 128.95, 128.75, 125.70, 125.51, 53.76, 31.40. HRMS: calculated for C_19_H_14_Cl_2_N_2_O_3_ [M − H]^−^: 387.0468, found: 386.9793.

*2-(5-(2-nitrobenzyl)-6-oxo-3-phenylpyridazin-1(6H)-yl)acetic acid* (**16**): Yellow solid, Yield = 80%; m.p = 102 °C (EtOH), FT-IR (ATR, cm^−1^): 3465 (OH), 3050–2865 (CH), 1756, 1650 (C=O), 1610 (C=N); ^1^H NMR (500 MHz, DMSO-*d*_6_, δ (ppm)) δ: 8.21 (t, *J* = 1.9 Hz, 1H, H-Ar), 8.13 (s, CH-pyri), 8.07–8.04 (m, 1H, H-Ar), 7.84–7.78 (m, 3H, H-Ar), 7.56 (t, *J* = 7.9 Hz, 1H, H-Ar), 7.49–7.41 (m, 3H, H-Ar), 4.87 (s, 2H, NCH_2_CO), 3.98 (s, 2H, CH_2_-Ar); ^13^C NMR (101 MHz, DMSO-*d*_6_, δ (ppm)) δ: 168.10, 159.75, 148.83, 144.30, 141.27, 140.85, 136.82, 134.86, 130.31, 130.17, 129.77, 129.13, 126.46, 124.44, 121.23, 54.65, 35.48. HRMS: calculated for C_19_H_15_N_3_O_5_ [M + 2H]^+^: 367.1073, found: 367.1427.

*2-(5-(4-fluorobenzyl)-6-oxo-3-phenylpyridazin-1(6H)-yl)acetic acid* (**17**): Brown solid, Yield = 86%, m.p = 187 °C, FT-IR (ATR, cm^−1^): 3460 (OH), 3085–2842 (CH), 1750, 1649 (C=O), 1598 (C=N); ^1^H NMR (500 MHz, DMSO-*d*_6_, δ (ppm)) δ: 7.68–7.55 (m, 5H, H-Ar), 7.48–7.40 (m, 4H, H-Ar), 7.04 (s, 1H, CH-pyr), 4.94 (s, 2H, NCH_2_CO), 3.20 (s, 2H, CH_2_-Ar); ^13^C NMR (101 MHz, DMSO-*d*_6_, δ (ppm)) δ: 168.85, 158.91, 143.24, 139.37, 135.35, 134.13, 132.51, 130.07, 129.43, 128.97, 128.76, 125.71, 125.52, 53.75, 31.40. HRMS: calculated for C_19_H_15_FN_2_O_3_ [M + H]^+^: 339.1101, found: 339.1997.

### 3.3. Theoretical Methods

Theoretical calculations are used to compare the biological and chemical activities of molecules through the determination of many quantum chemical parameters and each calculated parameter gives information about the different properties of the molecule. Gaussian09 RevD.01, GaussView 6.0 [21,49,50] was used to calculate these parameters. DFT calculations with B3LYP/6-31++g(d,p) basis set were performed [22,51]. Many parameters were calculated, namely HOMO, LUMO, ΔE (HOMO-LUMO energy gap), electrophilicity (*ω*), chemical potential (*μ*), bulk softness (*σ*), chemical are hardness (*η*), nucleophilicity (*ε*), energy value, and dipole moment [51,52,53]. These parameters were calculated using the following equations.
$$\chi =-{\left(\frac{\partial Ε}{\partial Ν}\right)}_{\upsilon \left(r\right)}=\frac{1}{2}\left(I+A\right)≅-\frac{1}{2}\left(E_{HOMO}+E_{LUMO}\right)$$
$$\eta =-{\left(\frac{\partial^{2}Ε}{\partial {Ν}^{2}}\right)}_{\upsilon \left(r\right)}=\frac{1}{2}\left(I-A\right)≅-\frac{1}{2}\left(E_{HOMO}-E_{LUMO}\right)$$
$$\sigma =1/\eta \;\;\;\;\;\;\;\;\;\omega =\chi^{2}/2\eta \;\;\;\;\;\;\;\;\;\;\;\;\;\;\varepsilon =1/\omega$$

In order to examine the antibacterial activities of the studied compounds, their inhibitory activities against various proteins were investigated. For this review, the Maestro molecular modeling platform (version 12.8) by Schrödinger [54] was used. For this calculation, the protein preparation module is used for the preparation of proteins [55], the LigPrep module [56] is used for the preparation of molecules, and the Glide ligand docking module [57] is used to interact between the prepared molecules and proteins. In the molecular-docking calculations, since the flexibility feature was added to the studied ligand molecules and the proteins in bacteria, both horizontal approach and vertical interactions were ensured. Moreover, energy minimization steps were performed using Macromodel (Schrodinger) and OPLS3e force field by following the Polak−Ribiere conjugate gradient (PRCG) algorithm with an energy gradient of 0.01 kcal/mol. However, the study was conducted using ADME/T analysis so that the studied molecules could be used as drugs. The effects and reactions of compounds in human metabolism were predicted by using the Qik-prop module of the Schrödinger Release 2021-3 (New York, NY, USA) [58].

### 3.4. Antibacterial Activity

The compounds (**3**–**17**) were evaluated in vitro for their antibacterial activities against *S. aureus* (methicillin resistant) ATCC 43300, *E. coli* ATCC 25922, *A. baumannii* ATCC 19606, *P. aeruginosa* ATCC 27853, and *S. typhimurium* ATCC 14028 bacterial strains. The antibacterial activities of the title compounds were determined by microdilution broth assay following the procedure described in our previous work [59].

## 4. Conclusions

In summary, novel pyridazinone derivatives (**3**–**17**) were synthesized, characterized and screened for their potential antibacterial activities against *S. aureus* (methicillin resistant), *E. coli*, *A. baumannii*, *P. aeruginosa*, and *S. typhimurium* bacterial strains. Most compounds showed significant antibacterial activities. Compounds **7** and **13** showed the most potent antibacterial activities against *S. aureus* (MRSA), *P. aeruginosa*, and *A. baumannii* with an MIC value range of 3.74–8.92 µM. DFT calculations with B3LYP/6-31++g(d,p) level were used to analyze the electronic and geometric characteristics deduced for the stable structure of title compounds and the principal quantum chemical descriptors were correlated with the antibacterial activity. However, it was found to have higher activity than other molecules with a docking score of −7.31 kcal/mol against *S. aureus* protein and −5.69 kcal/mol against *P. aeruginosa* protein. In addition, the predicted ADMET profiles of some derivatives were in line with Lipinski rules. As a result, this study will be an important guide for future in vivo studies and these new structures can be considered interesting for further modification as antibacterial agents.

## Data Availability

Not applicable.

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
