# Peer review of "Synthesis, Spectroscopic Characterization, Antibacterial Activity, and Computational Studies of Novel Pyridazinone Derivatives"

_molecules, 2023, doi:10.3390/molecules28020678_

Round 1

Reviewer 1 Report

The manuscript entitled "Synthesis, spectroscopic characterization, antibacterial activity and computational studies of novel pyridazinone derivatives" by Daoui et al involved the interesting research of 15 novel molecules and their antibacterial activity analyses with the corresponding theoretical approaches to understand these resultas.

The introduction is according to the developed topic of the manuscript, and it has updated bibliographical references to support the research.

Additionally, the manuscript is clear, organize, and easy to follow according to the objective of the authors.

However, there are some points to considering:

1)             It is necessary to clarify the importance of this kind of pyridazinone structures in order to understand the importance of this topic or why the authors have selected them (introduction part). Furthermore, why the authors only use H, CH3, F, Cl, and NO2 groups to analyze the potential bioactivity of the synthesized molecules? It is necessary to to clarify this point.

2)             Considering the chemical/synthetic nature of this manuscript it is necessary to discuss the structural differences among the selected compounds and their biological activities (not only report the results or discuss the information in relationship to the computational analyses) to understand the importance of the chemical groups and structure difference among them. Moreover, the difference between series (8-12) and series (13-17) is the carboxylic acid after the hydrolysis step. Did the authors analyse the pH of the biological assay medium in order to relate the results with the computational studies? Also, regarding the use of these specific bacteria strains, could the authors discuss the results according to the nature of them? (Gram + and Gram -; they have different structures).

3)             Did the authors check the synthesized molecules didn´t present PAINs (for the biological assays results)?

4)             Line 379: the authors reported that “As a result of the examination, it is seen that the studied molecules can be used as drugs in human metabolism”. It should be necessary to perform additional bioassays in order to confirm this asseveration. Moreover, the authors didn´t analyze the cell viability of these compounds (cytotoxicity assays).

Furthermore, I encourage the authors to check some mistakes such as:

- Line 66: in vitro should be in italics

- Lines 73, 76, 81: pyridazin-3(2H)-one, the H atoms should be in italics (nomenclature)

- Please remember to use a blank space between the number and magnitude: 20 °C, 6 h, etc. Please check them all throughout the manuscript.

- Table 2. Please clarify the meaning of NA (entry 16)

- Figures 2,3, and 4. Please improve the quality of t he images interactions (they seem blurry)

- Line 205. Did the authors use dry ethanol for this step?

- Line 247. Did the authors use dry THF for this step?

Finally, I would like to invite the authors to include the abbreviation list of words at the end of this manuscript.

Reviewer 2 Report

The authors synthesized and made in silico and in vitro assays to a family of pyridazinone derivatives. The suggestion for their work is presented as follows:

  1. Line 29, the hybrid functionals are badly written in the text as “B3lyp and M062X”. The correct form is B3LYP and M06-2x. Also, the basis set needs to be corrected, as 3-21G, instead of 3-21g. In line 62, the authors write a double polarized and diffuse basis set, which is not mentioned in the abstract section.
  2. Line 96, the term MICs is not defined. 
  3. The results regarding the conceptual DFT are not discussed, only an explanation of what each quantum descriptor means.
  4. Figure 2-4 is poorly presented; it is not correctly labeled (for example, Figure 2A and Figure 2B), and the text below the left images is very small and hard to read.
  5. The units in table 3 are not specified. 
  6. In lines 347-350 is not explained if the descriptors were computed using a vertical approach or an adiabatic approximation.
  7. The manuscript does not mention the scoring function used in the docking assay, and the information about the search algorithm is lacking.
  8. Conclusions do not match the results, and the theoretical DFT calculations are not explained and contrasted with the experimental results.

Round 2

Reviewer 2 Report

The authors follow the recommendations, but it is necessary to specify if the quantum parameters are computed based on a single point (vertical parameter) or using the system's relaxation.